# Factors That Influence Maternal Child Health Nurses’ Identification of Risk of Family Violence to First Nations Women in Australia

**DOI:** 10.3390/ijerph22020217

**Published:** 2025-02-04

**Authors:** Catherine Louise Austin

**Affiliations:** Institute of Health and Wellbeing, School of Nursing and Midwifery, Mt. Helen Campus, Federation University Australia, Ballarat 3350, Australia; catherineaustinconsulting@gmail.com

**Keywords:** violence, maternal child health nursing, continuity of care, First Nations women

## Abstract

**Aim:** To understand the factors that influence family violence towards First Nations women, to inform practises and policies to support these women and improve their engagement in maternal child health services. **Design:** A qualitative study, using narrative inquiry integrated with the Indigenous philosophy ‘Dadirri’, and thematic analysis of the data. **Materials and Methods:** Survey of 10 Maternal Child Health nurses in 2019, and interviews of 35 Aboriginal mothers in 2021. **Results:** The nurses identified drugs, alcohol, socio-economic issues, the history of effects of colonisation on First Nations peoples, and stress as perceived factors influencing family violence, and acceptance, fear, cultural beliefs, and mistrust, for women’s low reporting of violence. Factors that influenced nurses’ ability to identify family violence were mistrust and understanding of Aboriginal culture. Low self-esteem, lack of belonging, and not being heard were identified by the mothers as factors that influence family violence. Fear of child protective services, shame, mistrust, and poor rapport with the nurses contributed to their low reporting of violence. The most significant factor for the mothers to disclose violence is fear of losing her child, mistrust, and the questioning process. **Conclusions:** Nurses’ understanding of Indigenous culture is critical to develop trust and improve the engagement of First Nations women. A significant difference in the synthesis of data between the nurses and their First Nations consumers was conspicuous. Research regarding the benefit of models and interventions that recognise the social determinants of health and well-being on health outcomes as well as the value of culturally strong health services aimed to encourage an earlier identification of risk, ideally from the antenatal period to the child’s fifth birthday, is imperative. The implications of this research are of international importance for First Nations families and challenge current nursing practises to address the human rights challenge of the inequity in health outcomes between First Nation and non-First Nation children, their exposure to family violence, and their over-representation in child protection services.

## 1. Introduction

The United Nations General Assembly 2013 recognises the rights of children to the highest attainable standard of health, focusing on preventative and primary healthcare for children, prenatal and postnatal healthcare for mothers, and reducing infant and child mortality [1]. Yet, in countries with a history of colonisation, such as Canada, the United States, New Zealand, and Australia, long-term poor health and well-being remain, burdening Indigenous families affected by the ongoing stressors of colonisation [2]. Additionally, access to services and resources for Indigenous peoples can be affected adversely by historical and contemporary social determinants of health [3]. These stressors can manifest in behaviours such as alcohol, smoking, and other drug misuse, psychological distress, mental illness, or violence, and can consequently influence their ability to care for their children [3]. There is no formal definition of “Indigenous peoples” in a global context, respecting the right of each Indigenous people to define themselves [1]. Indigenous peoples are distinct cultural groups that share ancestral ties to the lands where they live. According to the World Health Organisation (WHO), the land on which they live is inextricably linked to their identities, cultures, livelihoods, and their physical as well as spiritual well-being [4].

The need for improved healthcare for Indigenous children is evident in the ongoing disparities in child health among Indigenous and tribal populations in these countries [2]. Access in the early years to appropriate and timely culturally strong, holistic, and effective community-based services that are integrated yet flexible to suit the needs of the family is a recognised predictor of the success of a child’s transition to school, education, employment outcomes, and long-term well-being [5]. This is paramount in the child’s first 2000 days (the period from conception to the child’s fifth year), as this period is the critical period of foetal and child development and forms the foundation for the child’s development and health through their lifetime [6]. The period of early childhood is also an opportunity, wherein high-quality health and educational interventions can reap benefits that extend across the life-course [4,5,6,7].

Maternal and Child Health (MCH) nurses in Victoria, Australia, referred to as ‘child and family nurses’ in some other Australian jurisdictions, influence the shape of this critical period in a child’s life. Engagement in the MCH service enables the opportunity to prevent, identify, and respond to the challenges faced by families with children aged from birth to five years [8,9]. Despite the aims of MCH service provision, some groups, namely culturally and linguistically diverse (CALD) and Indigenous communities, either do not engage with services [10] or do not sustain engagement [11], and families experiencing disadvantage are less inclined to access services [12], bringing into effect an inverse care law in which the families who most need intense, high-quality care are least likely to receive it [12].

However, the impacts of colonisation have resulted in a distrust of government agencies, contributing to ongoing dilemmas in Australia’s Indigenous peoples’ health [13]. The Indigenous population in Australia are referred to as ‘First Nations’, ‘Aboriginal and Torres Strait Islanders’, or ‘Aboriginals’ if the population does not include peoples from the Torres Strait Islands of Australia [14]. In the last ten years, governments have attempted to improve outcomes for First Nations peoples by implementing measures under the ‘Closing the Gap’ framework [15]. Despite this impetus, figures still show that almost fifty per cent of First Nations children in Victoria are identified as experiencing vulnerability, which is twice that of non-First Nations children [16,17,18].

As access to appropriate healthcare in the child’s first 2000 days is vital for the long-term health and well-being outcomes of both the mother and her child [19], it is vital that service models that provide healthcare, like the MCH service, promote and support access and engagement of all families. This will subsequently encourage identification of risk, such as family violence (FV), to the mother or her child, and the need for follow-up assessment, intervention, or referral and support, ideally from the antenatal period to the child’s fifth birthday (the first 2000 days). The WHO gave international significance to the epidemic rates of intimate partner violence (IPV) in 2002 by identifying it as a major public health concern for countries around the world [20]. The degree of violence against women in Australia and around the world is unknown, as the estimates are widely considered to underestimate the extent of the problem [21]. In Australia in 2014–2015, 2800 women were hospitalised after being assaulted by a spouse, and 55 women were killed by a current or former partner, equating to almost one woman each week [22]. Estimates in 2015 report that over 400,000 women in Australia had endured violence by a partner during pregnancy [22], and nearly 750,000 women had children in their care when they endured this violence by a former partner. Shockingly, 78 per cent of these women said that their children witnessed the violence [22].

Victoria’s Family Violence Protection Act defines FV as “behaviour by a person that causes a child to hear, witness or otherwise be exposed to the effects of violence (for example, property damage or a distressed family member) after the violence has occurred” [23] (p. 16). The Aboriginal and Torres Strait Islander definition of FV includes “physical, emotional, sexual, social, spiritual, cultural, psychological and economic abuse and can occur within families, intimate relationships, extended families, kinship networks and communities” [24] (p. 2). Hence, this paper will encapsulate all forms of violence, including IPV, under the term ‘family violence’ (FV).

According to Lievore [25], Aboriginal and Torres Strait Islander women residing in regional and remote areas experience rates of FV up to 45 times more than non-Aboriginal women do, and 1.5 times more than Aboriginal and Torres Strait Islander women residing in metropolitan areas. Violence is known to be a major cause of physical and mental illness among women of childbearing age [26] and is particularly common when women are pregnant or have recently given birth [21,26,27,28]. Additionally, if a woman has a history of FV in childhood, their trauma may be ‘triggered’ in the perinatal period (pregnancy to two years postpartum) and may seriously impact on how they raise their child, ultimately affecting the development of that child [25,27,29,30]. This, in turn, can lead to long-lasting relational effects and intergenerational cycles of trauma [29]. Smith argues that children exposed to FV could encounter developmental, emotional, and behavioural problems and long-term issues with social competence [27].

Over ninety-five per cent of mothers with newborns participate in the MCH service in Victoria, Australia [26], placing MCH nurses in the forefront to identify the risk of FV to these women and their children. Women are referred to MCH services by maternity services after birth, and a schedule of routine monitoring of child development, parenting support, and health promotion services is offered from the child’s birth to five years of age. MCH nurses in Victoria are Registered Nurses and Registered Midwives with a postgraduate qualification in Child and Family Nursing. Routine FV screening by Victorian MCH nurses at the four-week postnatal Key Age Stage (KAS) consultation was introduced in 2009, in response to research by the National Health and Medical Research Council (NHMRC) in Australia [31]. However, results from evaluation of the KAS framework found that FV screening was not implemented according to the recommendations, as the MCH nurses lacked confidence in the process [32]. According to the 2017–2018 MCH services Annual Report, 21,150 women, comprising 267 Aboriginal and Torres Strait Islander women and 20,883 non-Aboriginal and Torres Strait Islander women, attended the four-week consultation with MCH nurses in the Southwestern Region of Victoria, but only 732 (3.46%) of mothers were counselled for FV [26]. Of these 732 women counselled for FV, only 114 (15.57%) were referred to FV services [26], revealing that the data reported by MCH services regarding the prevalence of FV in women with children aged zero to five years may not accurately reflect the extent of the problem.

## 2. Materials and Methods

This research study is situated within a larger research study, which is investigating the development of a service model that is culturally sensitive, and effective in supporting the access and engagement of First Nations women in MCH services.

### 2.1. Aims

The aim of this smaller research component is to understand the factors that influence FV towards First Nations women with children aged zero to five years.

### 2.2. Research Questions

Q1. What factors influence family violence to First Nations women with children aged zero to five years?

Q2. What factors influence low reporting of family violence by First Nations women with children aged zero to five years to Maternal and Child Health nurses?

Q3. What factors influence Maternal Child Health nurses’ identification of risk of family violence to First Nations women with children aged zero to five years?

### 2.3. Design

The researcher has worked with a diversity of First Nations peoples across Australia (as a MCH nurse and as an Adviser to the Federal Minister of Aboriginal Australians) and is aware of the significant issues that need to be considered when conducting research with First Nations people, including any cultural sensitivities; perceived power imbalances between the researcher/s and First Nations participants; and issues of trust between the researcher/s and First Nations participants. Consequently, the researcher used the methodological principles of narrative inquiry integrated with the Indigenous philosophy Dadirri to achieve the research aims and objectives. Narrative inquiry is an interpretative approach from the social sciences that examines human lives through the lens of a ‘narrative’, or ‘storyline’ [33]. The use of narrative inquiry allowed the researcher to bridge Western and Indigenous research methodologies, providing a methodological approach of holistic observation from an Indigenous standpoint, without the risk of bias, to challenge and change thinking, ideas, and understanding [34] and to be congruent and in tune with the physical, psychological, social, and cultural aspects of the storytelling or ‘yarning’ of the First Nations women. Dadirri is a practice of “deep listening and acceptance” [35] (p. 4). This philosophy, used as a research method, enables Indigenous “voices to be heard” [35] (p. 4). Deep listening is a process of “listening to learn” [36] (p. 3). “It requires the temporary suspension of judgment, and a willingness to receive new information—whether pleasant, unpleasant, or neutral. This design facilitates two-way communication and creates a more democratic environment, whereby people can share their thoughts, ideas and opinions, regardless of corporate hierarchy, and improve relationships right across the structure. Effective two-way communication is required to solve problems better” [36] (p. 3).

The author recognises that Indigenous pedagogy is poorly understood and is not well represented in the education agenda [37]. As a non-Indigenous researcher, with vast experience working in collaboration with Indigenous communities in Victoria, the author’s site of struggle as depicted by Nakata, is “the need to change the idea of non-Indigenous researchers as the experts and to give Indigenous people a strong voice in all parts of research so that it can help to transform the lives of Indigenous people” [38] (p. 15). Pelto and Pelto add, “Indigenous people now want research and its designs to contribute to the self-determination and liberation struggles as defined and controlled by their communities. The inability of academia to acknowledge the differences between the two cultures can have a direct effect on the research outcomes and the difference between the emic and the etic data is the applied cultural sensitivity of the researcher” [39] (p. 6).

### 2.4. Setting

The study was conducted in the Glenelg Shire, a rural region in Southwestern Victoria (Figure 1). This region was selected due to the high population per capita of Aboriginal families [16]. Additionally, the reported rate of FV to Aboriginal women in this region was 39 per cent higher than non-Aboriginal women in this region [16,26]. As none of these women identify as ‘Torres Strait Islanders’, the population in this setting will be referred to as ‘Aboriginal’, ‘First Nations’, or ‘Indigenous’, rather than ‘Aboriginal and Torres Strait Islanders’. Some of the participants in the study refer to the colloquial term ‘mob’, which identifies Aboriginal people from a particular place or country and can refer to a family, clan, or community group [40].

### 2.5. Sample/Participants

In this study, the stratified purposeful sampling of two unique data sources was collected.

#### 2.5.1. Maternal and Child Health Nurses

To understand what MCH nurses perceive as factors that influence FV to First Nations women with children aged zero to five years, and the factors that influence MCH nurses’ identification of risk of this violence, the author sought to recruit MCH nurses with clinical experience engaging First Nations women and their children in MCH services. The inclusion criteria for this data source were MCH nurses employed to work as an MCH nurse in the Glenelg Shire, Victoria, Australia. The full population of MCH nurses in this setting were invited to participate in the study.

#### 2.5.2. First Nations Women with Children Aged Zero to Five Years

To understand the experiences and perspectives of First Nations women and what they identify as factors that influence FV to Indigenous women with children aged zero to five years, and the factors that influence MCH nurses’ identification of risk of this violence, the author sought to recruit First Nations women with children aged birth to zero years. The inclusion criteria for this data source were all First Nations women with children aged zero to five years residing in the Glenelg Shire, Victoria, Australia. Stratified purposeful sampling was employed to recruit three unique sub-categories of participants, as defined by their level of engagement in MCH services. These three levels of engagement include current engagement; initial engagement, now disengagement; and no engagement in the MCH service in the Glenelg Shire, Victoria, Australia. The sample size was largely determined by the number of respondents available to participate in the study. Based on the literature [41,42], evidence-based recommendations for the a priori estimation of sample sizes for each of the categories and sub-categories were seven–ten participants. The author estimated that data saturation would be achieved by recruiting ten participants in each category and sub-category. The author deemed a target of seven to ten participants per category and subcategory as appropriate due to the exploratory nature of the research and the focus on identifying underlying ideas about the topic. The author also anticipated difficulty accessing and engaging the First Nations women in the study.

### 2.6. Data Collection

Although the sample was relatively small for each category and sub-category, the material collected was quite detailed given the focused nature of the study, so the author felt the number was sufficient to satisfy the aims of this small in-depth investigation [43].

#### 2.6.1. Maternal and Child Health Nurses

The author consulted with the Victorian Department of Health and Human Services (DHHS), the Victorian Department of Education (DET), and the Municipal Association of Victoria (MAV) for ethics approval to collect data from MCH nurses employed in the Glenelg Shire. Once ethics approval to conduct this research was obtained (project number 2014_002311), a qualitative survey using open-ended questions to individualise the responses and reveal the opinions and experiences of the MCH nurses and to find the answers to the synopsis of the study question was developed. The survey instrument was assessed by an expert panel of MCH nurses with experience in qualitative research. MCH nurses, independent of the research and those who were not employed in the Glenelg Shire (n = 6), were recruited for pilot testing to ensure the validity of the study instruments in the survey research. Valid suggestions from this pilot testing were incorporated into the survey design and some minor clarifications were made to the questions. Following ethics approval, the survey was then distributed to all MCH nurses (n = 10) employed in Glenelg Shire, Victoria, Australia, between August and November 2019. A Plain Language Information Statement (PLIS) outlining the project and a consent to participate in the research form was provided to participants, explaining the purpose and intent of the project and how the data were to be collected and used. All MCH nurses completed the survey, which equated to 100% of the sample size (n = 10).

#### 2.6.2. First Nations Women with Children Aged Zero to Five Years

The author consulted with the relevant agencies for ethics approval to collect data from individual and small groups of First Nations mothers with at least one child aged zero to five years, residing in the Glenelg Shire These agencies included the Victorian DHHS, the Victorian Department of Health (DH), the MAV, the Dhauwurd Wurrung Elderly and Community Health (DWECH) Aboriginal Community Controlled Health Organisation (ACCHO), Winda-Mara Aboriginal Corporation ACCHO, and the Victorian Aboriginal Community Controlled Health Organisation (VACCHO).

Once ethics approval to conduct the research was obtained (project number C17-024), the author consulted with traditional owners residing in the two ACCHOs to gain assistance with the development of the research guide and interview questions, and to ensure the research plan was culturally appropriate. A panel of well-informed or experienced stakeholders and Indigenous knowledge holders was then consulted to clarify the relevance and clarity of each question/discussion prompt on the indicative interview schedule and to establish face validity. This panel comprised the research project team, an experienced MCH nurse researcher, and key representatives from VACCHO and the ACCHO sites. A small number of the target group, independent of the research, piloted the questions. Any valid suggestions from the expert panel and pilot testing were incorporated into the interview schedule design and clarifications were made to the questions/prompts where appropriate.

Following full approval of the research, the Chief Executive Officers (CEO’s) of the two ACCHO’s were provided with an overview of the project and timeframes, and a First Nations employee within each ACCHO was appointed by the CEO as a ‘site coordinator’ to act as the chief point of contact with the project team and assist with recruitment of discussion participants. Women who met the inclusion criteria were then invited to participate in the research (n = 47). Six of these First Nations women resided in the far northern part of the Glenelg Shire and did not engage with the two nominated ACHHO sites, so were excluded from the study (n = 41). A Plain Language Information Statement (PLIS) outlining the project and a consent to participate in the research form was provided to participants, explaining the purpose and intent of the project and how the data were to be collected and used.

The author consulted with the Aboriginal women face-to-face at the DWECH Service and the Winda-Mara Aboriginal Corporation ACCHOs in December 2021, using in-depth semi-structured discussion (‘yarning’). The discussions ran for approximately an hour at the ACCHO sites, co-facilitated by key staff from the ACCHOs. Of the 25 women that met the inclusion criteria in Portland, 84.0% (n = 21) participated in the study facilitated by the DWECH Service in Portland. Two venues were chosen, at the recommendation of the site coordinator, to increase the recruitment of participants, as some mothers engaged in a particular programme over another due to ‘bad blood’ at the alternative venue. Eight women were interviewed at the DWECH Service Playgroup (Site A, n = 8) and thirteen at the DWECH Service Women’s Group (Site B, n = 13). Of the 21 women interviewed in Portland (Site A+B), 28.6% (n = 6) had never engaged in the MCH service, 42.8% (n = 9) had disengaged in the MCH service, and 28.6% (n = 6) were currently engaged in the MCH service. All women (n = 21, 100%) identified as ‘Aboriginal’. Of the 16 women that met the inclusion criteria in Heywood, 87.5% (n = 14) participated in the study at the Winda-Mara Aboriginal Corporation ACCHO in Heywood, Victoria, Australia (Site C). All women (n = 14, 100%) identified as ‘Aboriginal’. Of the 14 women interviewed in Heywood, 28.6% (n = 4) had never engaged in the MCH service, 14.3% (n = 2) had disengaged in the MCH service, and 57.1% (n = 8) were currently engaged in MCH services.

### 2.7. Data Analysis

The audio-recorded data were transcribed by the author and subjected to attributional, first, and second cycle coding. Data were analysed by the author using Braun and Clarke’s six-step process for identifying, analysing, and reporting qualitative research using thematic analysis [44]. Thematic analysis facilitated the emergence of themes and patterns to identify broad concepts of the barriers and enabling factors that influence MCH nurses’ identification of risk of FV to First Nations women in Australia. This process included familiarisation with the data, and then generating initial codes, searching for themes, reviewing the themes, defining and naming the themes, and producing a report with the themes found within the data. A key First Nations academic with experience in conducting research with First Nations mothers in ‘the first 1000 days’ of their child’s life, and the two site coordinators, were involved in the analysis of the data to ensure there was an Indigenous lens applied to the thematic analysis of the data.

### 2.8. Validity and Reliability/Rigour

Research validity was ensured through consultation with a panel of well-informed and/or experienced stakeholders and Indigenous knowledge holders who reviewed the questions for the survey and interviews, and pilot testing of the survey and interviews prior to initiation was used to identify any misunderstandings or inaccuracies in the questions [45]. Consequently, this provided the author with the opportunity to implement changes to the survey and interview questions resulting from any feedback from the expert panel and pilot.

## 3. Results

The MCH nurses and the First Nations women were asked their perspectives on the same three questions, with the aim to understand the factors that influence MCH nurses’ identification of risk of FV to First Nations women with children aged zero to five years. Coded interpretation of the individual data items within the dataset were analysed into themes and sub-themes that focus on the difference in responses between the MCH nurses and the First Nations women, to represent the over-arching narrative within the data to help answer the research questions.

### 3.1. Factors Contributing to Family Violence to First Nations Women with Children Aged Zero to Five Years

To understand the factors that influence FV, the author asked the MCH nurses and the First Nations women for their perception or experience of factors that influence the risk of FV to First Nations women with children aged zero to five years. All of the nurses answered this question (n = 10, 100%); however, not all of the Aboriginal women chose to (n = 12, 34.3%), citing that “it was a very awkward thing to discuss” (Participant C14). From the responses (see Figure 2), it was evident that half of the MCH nurses perceived that contributing factors to FV were drugs and alcohol (n = 5, 50.0%). Two nurses (n = 2, 20.0%) stated that they believed socioeconomic factors, such as education, employment, finances, money, and poverty, influence the risk of FV to First Nations women. The remaining MCH nurses believed that family reasons (n = 1, 10.0%), which included the woman’s partner, family, baby, gender, and relationships; history (n = 1, 10.0%), referring to a history of violence in the family to the extent that it had become an accepted and normalised behaviour that the family then perpetrates; and stress and/or mental health (n = 1, 10.0%) were contributing factors. In contrast, from the 12 Aboriginal women that answered this question, the majority (n = 7, 58.3%) said that a First Nations woman’s “low self-esteem” was the main contributing factor that influences the risk of FV to them, with one participant stating, “we have had so much taken away from us” (Participant PB1). Other factors were “lack of belonging” (n = 3, 25.0%) and “not being heard, no voice” (n = 2, 16.7%).

The nurses’ perception of factors that influence the risk of FV to the First Nations women were in stark contrast to the experiences of the First Nations women who participated in the research. The low response rate to this question by these women is a significant finding in this research, as over half of the women who did respond cited their poor self-esteem as a major factor contributing to FV. The poor insight of the MCH nurses into factors contributing to FV for the First Nations women and the resistance of these women to discuss this with MCH nurses will inevitably lead to poor identification of their risk of FV.

### 3.2. Factors That Contribute to Low Reporting of Family Violence by First Nations Women with Children Aged Zero to Five Years

To understand factors that contribute to Aboriginal women with children aged zero to five years disclosing their risk of FV, the author asked the MCH nurses and the First Nations women for their perception or experience of factors that influence the low reporting to MCH nurses of FV by First Nations women with children aged zero to five years. All participants answered this question. From the responses (see Figure 3), a little less than half of the MCH nurses perceived that a contributing factor was ‘acceptance’ (n = 4, 40.0%), which was related to the belief that MCH nurses felt that FV was perceived as being ‘normal’ in Aboriginal culture. Other nurses perceived that reasons for the low reporting of FV were fear (n = 3, 30.0%) and cultural beliefs (n = 2, 20.0%). Fear referred to the fear these families felt towards DHHS, child protection, and anyone in perceived ‘authority’, including MCH nurses. One nurse perceived that mistrust of the MCH service (n = 1, 10.0%) was a contributing factor, which was related to the category of ‘fear’ because MCH nurses are mandated to report any child protection issues to the necessary authorities. Overwhelmingly, the Aboriginal women cited ‘fear’ as a factor influencing the low reporting of FV to MCH nurses (n = 27, 77.1%). Their fear was in relation to their partner, authorities (such as child protection agencies, including MCH services), community, and family. One participant stated, “if we say to a white woman, we are going through DV, our anxiety of having our child/children removed is so high that we are not going to own up to it” (Participant PC6). Another said, “The Stolen Generation is still fresh in our minds, these are not stories from a long time ago. These things have happened to our grandmothers. Our grandmothers’ stories are whirling around in our minds” (Participant PB2). Many participants agreed they were ‘scared of child protection’, one participant saying that “we are scared to get the fathers in trouble and that our children will be taken away” (Participant PB10). Poor rapport/not having developed a relationship with MCH nurses to disclose this was cited as a cause in 11.4 % (n = 4) of the responses. One participant disclosed, “Mob do not like the authoritive tone when you speak to us. I felt judged so I didn’t talk” (Participant PA3). Another participant said, “as an Aboriginal woman we have all these things in your mind when you go and see a service, whether it’s FV or our partners or how we raise our children. We are always really listening and thinking as to how to answer the question. I didn’t go through FV, but I still felt nervous answering that question. You tense up and feel nervous about it” (Participant B5). One woman divulged, “as an Aboriginal woman, people think the worst straight away” (Participant PA2). A little less than 10% of the Aboriginal women cited mistrust of MCH nurses (n = 3, 8.6%) as a contributing factor, one participant stating, “Mob needs safety to tell her story, the trust isn’t there” (Participant PA4). Another participant said, “Maternal and Child Health nurses in Victoria are mandated to report risk to our children, so why would we say anything” (Participant PC3)? One mother stated, “I didn’t feel that my story was being heard so I didn’t trust the nurse” (Participant PB6). Another mother said that a contributing factor for her was shame (n = 1, 2.9%), which was about fear of judgement by MCH nurses, embarrassment, and poor self-esteem. This participant disclosed, “I don’t want the nurse to judge me and see me differently if I tell her my stories” (Participant PB12).

The responses from the First Nations women and the MCH nurses were more assimilated when asked about factors that contribute to the low reporting of FV by First Nations women with children aged zero to five years. The nurses accurately predicted the ‘fear’ and ‘mistrust’ that many First Nations women feel towards MCH nurses and their services as significant factors to their low reporting of risk of FV, but the MCH nurses did not identify that the strength of a woman’s relationship with them was a significant factor in the woman reporting her risk of FV.

### 3.3. Factors That Influence Maternal Child Health Nurses’ Identification of Risk of Family Violence to First Nations Women with Children Aged Zero to Five Years

The author then asked the MCH nurses and the First Nations women of their experiences and perceptions of factors that influence MCH nurses’ identification of risk of FV to First Nations women with children aged zero to five years. All participants answered this question. The responses (see Figure 4) indicate that nearly half of the MCH nurses (n = 4, 40.0%) identified the ‘process of the questioning’, which referred to the ‘inadequate training’ the MCH nurses stated they had to be ‘confident to ask about FV’, and not having the right time and/or place to ask the questions about the woman’s risk of FV. The same number of nurses (n = 4, 40.0%) felt that ‘gaining trust’ with the woman through continuity of care from Koori Maternity Services (KMSs) and other services attended by the woman and her family was a factor that influences MCH nurses’ identification of risk of FV to First Nations women with children aged zero to five years. One nurse (n = 1, 10%) thought that her poor understanding of Indigenous culture and inability to provide a service that was culturally sensitive to these families was as a significant factor. Another nurse said that she needed to be more aware of FV issues for women in their community (n = 1, 10%). Nearly half of the Aboriginal women (n = 15, 42.9%) said it was the questions MCH nurses ask, particularly the ‘tone’, ‘delivery’, or ‘wording’ of these questions, that are factors that influence identification of risk of FV to First Nations women with children aged zero to five years attending the MCH service. One participant said, “Don’t make it so white. It’s not what you say, it’s how you say it. You and I can say the same thing, but it could come out totally different” (Participant PB1). Another mother said, “It’s the tone, it’s the way it’s delivered, and not using that word family violence” (Participant PB2). Another participant said, “use a strengths-based approach when you speak to us” (Participant PB3) and another said, “ask better, like how is your well-being” (Participant PB4). One mother suggested to “re-phase the question if you don’t get a response” (Participant PB5). Another participant recommended to “steer away from what a child protection worker would say to an Aboriginal person and use more culturally appropriate language” (Participant PB7). Similarly, another mother said, “better conversation: the word ‘risk’ is associated with Child Protective Services” (Participant PC1). Another stated, “ask ACCHO’s how could we ask this better: this is how you should be asking us” (Participant PB9). Several First Nations women (n = 6, 17.1%) said that a better understanding of culture would influence MCH nurses’ identification of risk of FV to First Nations women with children aged zero to five years. One participant stated, “some mob prefer to have the nurse visit at home, but our homes are set up differently, we have lots of extended family in our homes. The nurses come in and see couch surfers and think that is bad, which can last for weeks and weeks and weeks. Seeing that one thing, they think crap, we must write this down, but it’s just my connection to family, that’s just kinship” (Participant PB10). Another said, “they judge us how they see it; they don’t ask us for our story” (Participant PB6). One participant suggested, to “stop stereotyping that domestic violence is a cultural experience, not everyone experiences it” (Participant PB11). Another agreed stating, “stop presuming that all Aboriginal women are experiencing domestic violence, because they are Aboriginal women! It is a perception that it IS happening” (Participant PC3). One participant suggested it might help to “have a MCH nurse based at the ACCHO, either permanently or a visiting one, or at least at other services such as playgroup or women’s group then they can pick up the way we do things and yarn about things, so they will be learning and passing onto their colleagues” (Participant PA7). Another significant factor to overcoming barriers to identifying FV, according to the women (n = 5, 14.3%), was better continuity of care between services. One participant said, “get a referral of the big picture from the Koori Maternity Services worker or the midwives who would have seen the mum for the whole nine months so would have a good idea if there is violence” (Participant PC7). Another stated, “better communication between services like the midwives so we don’t have to tell our story over and over again” (Participant PA8). One mother added, “MCH nurses could attend the GP ATSI health checks so there is not an overlap” (Participant PB13). Establishing a trustful relationship (n = 4, 11.4%) was also a factor. One mother said, “If I disclose domestic violence, I want reassurance from the MCH nurse that I won’t lose my support for my new baby” (Participant PA6). Another participant said MCH nurses needed to “establish a strong respectful relationship before asking such a personal question” (Participant PC10). One mother suggested to “attend sister days or women’s groups at ACCHO’s to get to know the women better and build trust” (Participant PA1). Another mother disclosed, “when the nurse visits a typical Aboriginal household there maybe toys on the floor and dishes in the sink from last week, and that’s when phone calls are made and questions aren’t asked, but my home is safe, but that’s how kids can be removed from that one incident. That’s how easy that is, the trust isn’t there” (Participant PA2). Another significant factor to overcoming barriers to identifying FV was the timing/setting of when the question was asked (n = 3, 8.6%). One participant said, “the MCH nurse should not ask whilst my partner is there” (Participant PA3). Similarly, another participant stated, “information given at the appointment for referral services if a mum can’t answer the question on the day would help” (Participant PA5). Another participant said, “I don’t know if this is possible, but if the nurse could say to the mum can we meet at the park or do you want to go for a walk or meet outside the home, women are going to feel more comfortable to disclose things if they are outside the home, other than where the violence is happening. You will learn so much when we are walking and yarning” (Participant PC11). Better communication about why the question is asked was also significant for a couple of mothers (n = 2, 5.7%). One participant said, “Mob get angry at being asked, not realising everyone gets asked” (Participant PA4). Another said, “better education to the community for what gets asked and checked at the key ages and stages and what happens at those appointments, displayed as artwork from a local artist” (Participant PB12).

The responses from the First Nations women and the MCH nurses to factors that influence identification of risk of FV to First Nations women with children aged zero to five years by MCH nurses were again quite comparable in this question. Issues around the ‘questioning’ of risk of FV was identified by nearly half of the nurses, and just over half of the First Nations women. Almost half of the MCH nurses also perceived that gaining trust with a First Nations woman was an important factor to enable them to identify the woman’s risk of FV, whereas only a small number of the First Nations women expressed this as a factor. Better understanding of localised Indigenous culture by the MCH nurse and improving the continuity of care between the KMS and MCH services were factors that were more significant for the women to disclose their risk of FV.

A coherent and lucid picture of the difference in responses between the MCH nurses and the First Nations women to represent the over-arching narrative within the data to help answer these research questions (see Figure 5) was evident in the coded data of the responses to the questions. Constituent data, presenting the same three concurrent narratives, was evident in Portland (Site A+B) and in Heywood (Site C). ‘Intercultural competence’ was clearly definable as a theme for the contrast in responses between the MCH nurses and the Aboriginal women in all three sites, depicting the First Nations mothers’ experience with discrimination and judgement as barriers to access MCH services. Intercultural competence refers to “the ability to demonstrate targeted knowledge, skills and attitudes that lead to effective and appropriate communication with people of other cultures” [46] (p. 261). The prefix ‘inter’ indicates the two-way interaction between individuals from two different cultures [46]. Concurrent narratives were constructed as separate sub-themes, namely ‘mistrust or fear’, ‘communication’, and ‘recognises the social determinates of health’, as common factors that influence MCH nurses’ ability to identify risk of FV to First Nations women with children aged zero to five years, and deterrents for the women to seek support for FV.

## 4. Discussion

Existing evidence shows that although First Nations people experience high rates of FV, their access to support is lower than their non-First Nations counterparts due to significant barriers including discrimination, shame, and fear [47]. It is also evident that there is poor health among First Nations families in countries with a history of colonisation, such as Australia, New Zealand, and Canada, and access to services is adversely affected by historical and contemporary social determinants of health, such as the distribution of power, influence, wealth, and income [2]. First Nations parents, in particular, require support services that are culturally strong, timely and appropriate, and holistic to strengthen their families’ health outcomes [8,9,48]. Ensuring the optimal design of a model that promotes and supports the engagement of First Nations families and their access to MCH services in the period from the child’s birth to five years of age is impeded by the dearth of information on the relationship between specific services and children’s health outcomes [4,49,50].

The findings of this study support existing evidence, including a review of the literature conducted in 2021 [51] by the author as part of a larger study, to identify models or interventions that promote and support better access and engagement, quality of care, service delivery, and outcomes for First Nations women and their children in MCH services. In the studies (n = 6) found in the literature review, enabling factors were service models or interventions that are timely, appropriate, culturally strong, effective, community-based services that are integrated but flexible to suit the holistic needs of the family. Barriers that impacted the access and engagement, quality of care, service delivery, and outcomes for First Nations women to MCH services were inefficient communication resulting in lack of understanding between client and provider, cultural differences between client and provider, poor continuity of care between services, lack of flexibility in approach or access to services, and a model that does not recognise the importance of the social determinants of health and well-being.

The findings also validate the Early Assessment Referral Links (EARL) concept, developed by the author in 2009 in collaboration with a broad cross-section of the Aboriginal community and other health service stakeholders in the Glenelg Shire, Victoria, Australia [52]. The aim of the EARL concept is to promote and support better access and engagement, quality of care, service delivery, and outcomes for Aboriginal women and their children in MCH services, and to identify families who require further assessment, intervention, referral, and/or support, ideally from the preconception or antenatal periods. The author evaluated the outcomes of 56 Aboriginal children and 30 Aboriginal women who participated in the trial of the EARL concept from 2009 to 2014. The participation of Aboriginal women in MCH services during the trial was consistently above the average participation rate across the State of Victoria in the same period, primarily as a significant proportion of Aboriginal women and their children were referred to EARL by professionals employed by First Nations services, or by other professionals who had an established rapport with the clients. Additionally, there were increases in Aboriginal children being breastfed, fully immunised, and attending preschool. The identification of First Nations women and children at risk of vulnerability also improved, with a dramatic increase in referrals for FV, from zero referrals before the trial commenced to fifteen referrals within a year of EARL being operational. A significant decrease in the number of children in Out of Home Care (OoHC) was also evident, from 37.93% (n = 11) of the total population of Aboriginal children aged zero to five years in Glenelg Shire residing in OoHC in the year before the EARL trial, to 4.25% (n = 2) in the second year of operation [52]. This trend remained after the trial was completed.

In addition, the findings of this study also support the results of the audit conducted by the Victorian Auditor General’s Office (VAGO) in 2014, which was established to examine the poor engagement of First Nations families in mainstream early childhood services, including MCH. The report from the audit [5] affirmed that the issue of accessibility of mainstream services for First Nations Victorians was multi-factorial, and included the quality-of-service standards, governance, policies, regulations, laws and acts, information sharing practises, data systems, and a skilled workforce [5]. Persistent barriers to Aboriginal families accessing MCH services in Victoria were identified by the audit as “a lack of culturally safe services; a lack of awareness of the services that are available; a lack of required services in the local area; racism; a lack of transport to service delivery; shame, embarrassment, fear” [53] (p. 7).

In response to the consistently lower participation rates of First Nations children compared to non-First Nations children at all 10 KAS consultations provided within the MCH service since the inception of the KAS model in 2009 [5], and to the findings from the VAGO audit (2016), DET initiated a review of engagement of First Nations families in MCH services. This was a two-phase review, with Phase 1 undertaken by VACCHO in late 2014. This phase examined the scope of MCH service provision to First Nations families and identified potential factors that affected access to MCH services by First Nations families from a local provider perspective [53]. Phase 2 of the DET review, which was conducted by ACIL Allen Consulting in 2015, built on the work of Phase 1 but sought to provide a broader understanding of the experiences of First Nations families with MCH services and to identify enablers and barriers to their access and engagement [50]. Both reports found that ACCHO-based MCH services had a stronger focus on social and cultural determinants of the health of First Nations people and were more flexible and tailored to the needs of First Nations children and families [53,54]. Co-location of support services provided within the ‘culturally safe environment’ of an ACCHO encouraged the holistic care of First Nations families and improved collaboration between the MCH service and other allied health services [53,54]. These findings, and those of the Royal Commission into Family Violence [15], concluded that there were ‘concerns’ with the differential response model framing early years services in Victoria to engage First Nations families. Due to recommendations and findings from VAGO [5], ACIL Allen [53], and wider, whole-government reforms, state-wide MCH nurse training on Trauma-Informed Practice, Risk Assessment and Management of Family Violence Risk, and Aboriginal Cultural Safety Training was implemented in Victoria.

However, the literature shows the lack of comparison to current knowledge gathered with First Nations women and non-First Nation service providers [39] and a conspicuous absence of First Nations women, children, and raising children in the context of FV [37]. The new knowledge that has emerged from the findings of this study, supported by the existing literature outlined in the discussion, reinforce the need to review the effectiveness of models of care to engage First Nations families to ensure that MCH services meet the needs of these families. Ensuring that First Nations women have an integral role in developing, implementing, monitoring, and evaluating plans and programmes for MCH, including the provision of FV services, would improve intercultural competence between the nurses and the women and support better access and engagement, quality of care, service delivery, and outcomes for these women and their children in MCH services.

### 4.1. Implications for Policy and Practice

The imbalance in the number of nurses in this study compared to the number of First Nations women in this study who recognise ‘fear’ as a deterrent for these women to disclose their risk of FV shows the stark reality of why the problem continues to exist. The MCH nurses themselves have an integral role in changing their practises and informing policies so that the whole sector is better prepared to address the fear that First Nations women are experiencing. To facilitate this, it is imperative that all MCH nurses understand First Nations culture and a have positive relationship within the First Nations community that they work in, including an authentic comprehension of their needs. Providing MCH nurses with training on how to integrate the principles of trauma- and violence-informed care into their practice might increase the physical, emotional, and cultural safety experienced by First Nations women in Australia. This would also ensure that MCH nurses provide a service based on best practice guidelines to monitor health and development and the prevention, early detection, and intervention of physical, emotional, and social factors, which may affect a child or their family [5,50,55]. To help identify the service needs of a First Nations community, MCH nurses need to ensure the accurate identification of First Nations children birth to five years of age by using reliable data collection and recording processes, and by ‘data matching’ with other organisations servicing this community. An accurate census of the population within a community will facilitate the better evaluation of programmes or interventions introduced with the aim to improve access and engagement, quality of care, service delivery, and outcomes for women and their children in MCH services and will identify gaps in service provision to prevent FV and protect these women and children. Outreaching to services that First Nations families use such as ACCHO’s, playgroups, kindergartens, and day-care facilities; attending Indigenous festivals/celebrations; and engaging with the women antenatally through collaboration with maternity services will encourage rapport and trust and will promote and support better access and engagement, quality of care, service delivery, and outcomes for First Nations women and their children in MCH services.

### 4.2. Limitations

There are three primary limitations in this study. The first is the reflexivity or potential effect of the author’s own background and life experiences and the influence this has on all aspects of the research process. The author is a white, middle-class woman. She is a midwife and a Maternal and Child Health nurse who works in the community where the research was conducted. This is potentially problematic as the author was the primary instrument of interpretation of the data, and the author’s prior experiences, assumptions and beliefs of class, gender, and race could potentially influence their interpretation of the data [56]. The author is aware of the potential bias that may have occurred when interpreting the data. To decrease this risk, the author took notes about the participants’ responses and the author’s thoughts during the interviews and analysed these data as soon as possible after the interviews. Second, due to the small number of participants, protecting anonymity may become an issue for those participants who do not want their stories shared. Third, the author predicts that the three sub-categories of First Nations women interviewed will represent a cross-section of Aboriginal women’s access and engagement in MCH services and may provide the insight to justify further research in the future with a greater number of participants. However, the author also acknowledges that this study does not have a good cross-section of geographical context, including representation of participants from metro, regional, and remote areas, disallowing geographical context being considered in the synthesis of the data.

## 5. Conclusions

The findings of this study show that there is a significant difference in the synthesis of evidence between the nurses and their First Nations consumers, prompting the need for further research, informed by Indigenous voices, regarding the benefit of alternative models and interventions that aim to encourage the prevention, or earlier identification, of risk. The data obtained in this study support the adequacy of the theory that a MCH model that supports a depository of best practice of Indigenous-informed services that are locally responsive and flexible to focus on client needs rather than the programme guidelines is an approach required to address the growing number of women and children reported to be at risk of significant harm. Most importantly, the findings of the study show that it is imperative that a MCH service model, like the EARL concept, should allow for the integration of traditional First Nations child-rearing practises with westernised values, practises, and beliefs through a guided mastery approach, shared knowledge, yarning, capacity building, mutual trust, and connection, to facilitate effective engagement and trust in the service. It is of international importance for First Nations families, nursing practises, and public health to address the human rights challenge of inequality in health outcomes between First Nations and non-First Nations children, the over-representation of these children in OoHC, and the high rates of their exposure to FV.

## Figures and Tables

**Figure 1 ijerph-22-00217-f001:**
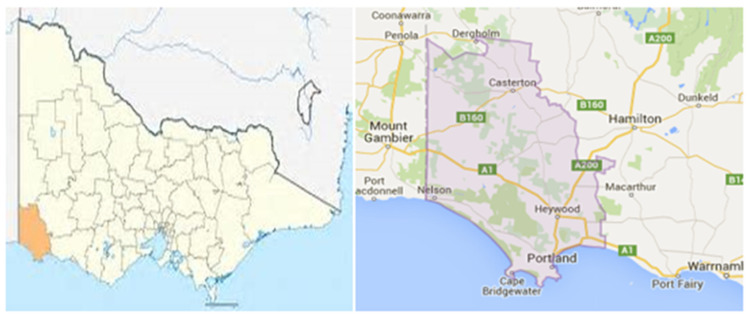
Glenelg Shire, South Western Victoria, Australia [26].

**Figure 2 ijerph-22-00217-f002:**
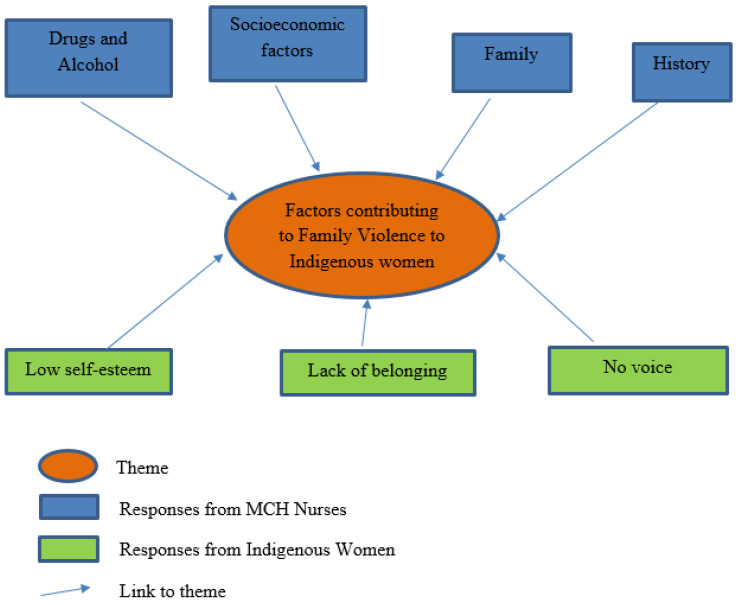
Factors contributing to family violence to First Nations women.

**Figure 3 ijerph-22-00217-f003:**
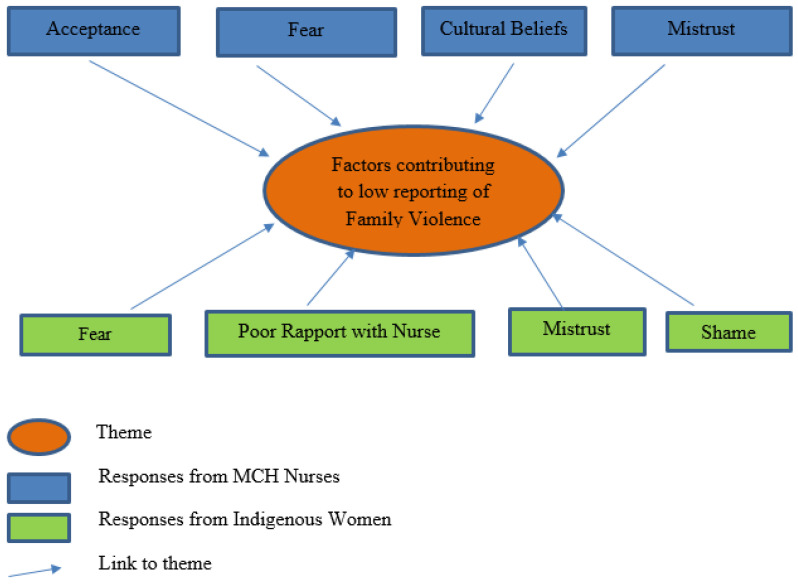
Factors contributing to the low reporting of family violence.

**Figure 4 ijerph-22-00217-f004:**
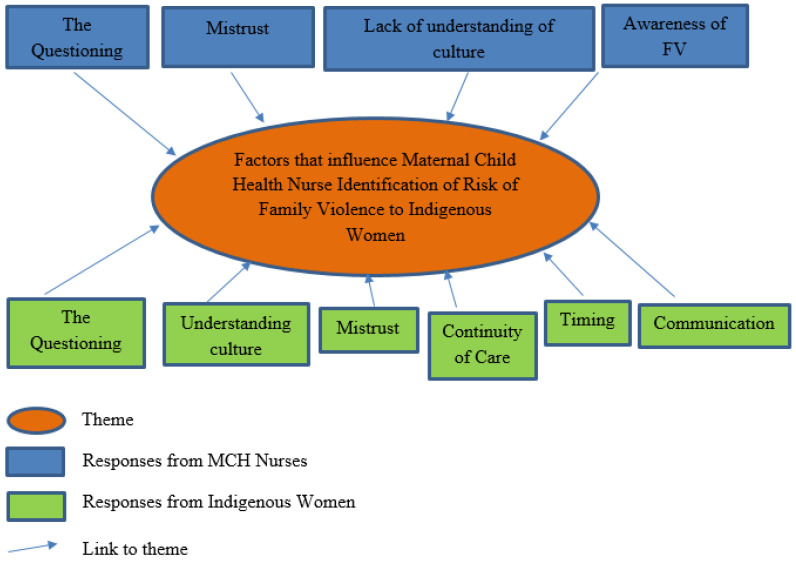
Factors that influence MCH nurses’ identification of risk of family violence.

**Figure 5 ijerph-22-00217-f005:**
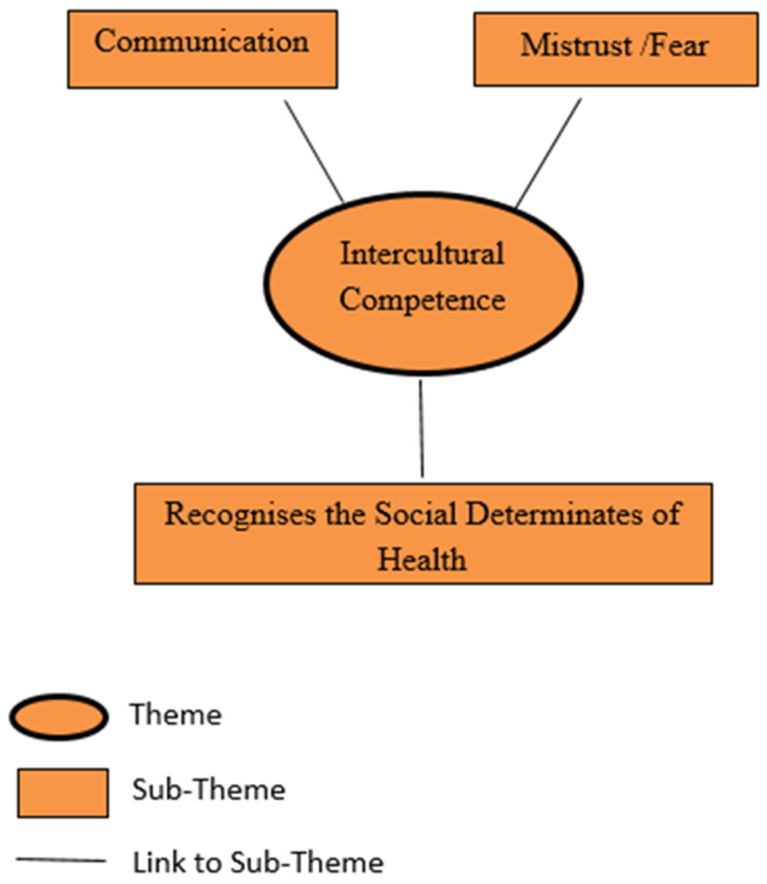
Finalised thematic framework map.

## Data Availability

The original contributions presented in this study are included in the article. Further inquiries can be directed to the corresponding author.

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
