# Peer review of "Factors That Influence Maternal Child Health Nurses’ Identification of Risk of Family Violence to First Nations Women in Australia"

_ijerph, 2025, doi:10.3390/ijerph22020217_

Round 1
Reviewer 1 Report
Comments and Suggestions for Authors
I congratulate you for the study, I read it with pleasure. A scientifically appropriate method was used. However, I think that more current resources can be used. Also, considering that qualitative research is insufficient in determining the relationship due to its nature, I think it would be more appropriate to give the word "relationship" instead of the word "determined themes".
Summary: As a result, you can write what you found in relation to the purpose and what is your result next to the suggestion
I also think that determining the 3rd title as "Matertal" or "Matertal and Method" would be more appropriate. Also, the purpose and research questions should be at the end of the introduction, not under the method.
I also think that there should be an ethics title (ethics committee and number) in the method section.
Author Response
Reviewer One: I congratulate you for the study, I read it with pleasure. A scientifically appropriate method was used. However, I think that more current resources can be used.
Author’s Response: Thank you for your positive feedback. I have reviewed the resources and updated them where appropriate/necessary which is evident in the track changes and the references in the reviewed/clean copy of the article.
Reviewer One: Also, considering that qualitative research is insufficient in determining the relationship due to its nature, I think it would be more appropriate to give the word "relationship" instead of the word "determined themes".
Author’s Response: As stated in the article, The audio-recorded data was transcribed by the author and subjected to attributional, first and second cycle coding. Data were analysed by the author using Braun & Clarke’s six-step process for identifying, analysing, and reporting qualitative research using thematic analysis [44]. Thematic analysis facilitated the emergence of themes and patterns, to identify broad concepts of the barriers and enabling factors that influence MCH nurses’ identification of risk of FV to First Nations women in Australia. This process included familiarisation with the data, then generating initial codes, searching for themes, reviewing the themes, defining and naming the themes, and producing a report with the themes found within the data. A key First Nations academic with experience in conducting research with First Nations mothers in ‘the first 1000 days’ of their child’s life, and the two site coordinators, were involved in the analysis of the data to ensure there was an Indigenous lens applied to the thematic analysis of the data. As a result, I have not accepted this recommendation.
Reviewer One: Summary: As a result, you can write what you found in relation to the purpose and what is your result next to the suggestion.
Author’s Response: I believe I have documented the results of the research in relation to the purpose and the implications for policy and practice in the original manuscript in line with Braun & Clarke’s six-step process for identifying, analysing, and reporting qualitative research using thematic analysis [44]. As a result, I have not accepted this recommendation.
Reviewer One: I also think that determining the 3rd title as "Matertal" or "Matertal and Method" would be more appropriate.
Author’s Response: Apologies, I don’t understand what you are suggesting here. I am presuming the word "Matertal", is meant to be “Maternal”? I can’t see where/what you are referring to. As a result, I have not accepted this recommendation.
Reviewer One: Also, the purpose and research questions should be at the end of the introduction, not under the method.
Author’s Response: I have been taught in my training in Australia to put the research questions where they are for this type of manuscript. I would prefer not to change where it is if that’s acceptable to you.
Reviewer One: I also think that there should be an ethics title (ethics committee and number) in the method section.
Author’s Response: Thanks for this feedback which I have accepted and edited the manuscript accordingly.Reviewer One: I congratulate you for the study, I read it with pleasure. A scientifically appropriate method was used. However, I think that more current resources can be used.
Author’s Response: Thank you for your positive feedback. I have reviewed the resources and updated them where appropriate/necessary which is evident in the track changes and the references in the reviewed/clean copy of the article.
Reviewer One: Also, considering that qualitative research is insufficient in determining the relationship due to its nature, I think it would be more appropriate to give the word "relationship" instead of the word "determined themes".
Author’s Response: As stated in the article, The audio-recorded data was transcribed by the author and subjected to attributional, first and second cycle coding. Data were analysed by the author using Braun & Clarke’s six-step process for identifying, analysing, and reporting qualitative research using thematic analysis [44]. Thematic analysis facilitated the emergence of themes and patterns, to identify broad concepts of the barriers and enabling factors that influence MCH nurses’ identification of risk of FV to First Nations women in Australia. This process included familiarisation with the data, then generating initial codes, searching for themes, reviewing the themes, defining and naming the themes, and producing a report with the themes found within the data. A key First Nations academic with experience in conducting research with First Nations mothers in ‘the first 1000 days’ of their child’s life, and the two site coordinators, were involved in the analysis of the data to ensure there was an Indigenous lens applied to the thematic analysis of the data. As a result, I have not accepted this recommendation.
Reviewer One: Summary: As a result, you can write what you found in relation to the purpose and what is your result next to the suggestion.
Author’s Response: I believe I have documented the results of the research in relation to the purpose and the implications for policy and practice in the original manuscript in line with Braun & Clarke’s six-step process for identifying, analysing, and reporting qualitative research using thematic analysis [44]. As a result, I have not accepted this recommendation.
Reviewer One: I also think that determining the 3rd title as "Matertal" or "Matertal and Method" would be more appropriate.
Author’s Response: Apologies, I don’t understand what you are suggesting here. I am presuming the word "Matertal", is meant to be “Maternal”? I can’t see where/what you are referring to. As a result, I have not accepted this recommendation.
Reviewer One: Also, the purpose and research questions should be at the end of the introduction, not under the method.
Author’s Response: I have been taught in my training in Australia to put the research questions where they are for this type of manuscript. I would prefer not to change where it is if that’s acceptable to you.
Reviewer One: I also think that there should be an ethics title (ethics committee and number) in the method section.
Author’s Response: Thanks for this feedback which I have accepted and edited the manuscript accordingly.Reviewer One: I congratulate you for the study, I read it with pleasure. A scientifically appropriate method was used. However, I think that more current resources can be used.
Author’s Response: Thank you for your positive feedback. I have reviewed the resources and updated them where appropriate/necessary which is evident in the track changes and the references in the reviewed/clean copy of the article.
Reviewer One: Also, considering that qualitative research is insufficient in determining the relationship due to its nature, I think it would be more appropriate to give the word "relationship" instead of the word "determined themes".
Author’s Response: As stated in the article, The audio-recorded data was transcribed by the author and subjected to attributional, first and second cycle coding. Data were analysed by the author using Braun & Clarke’s six-step process for identifying, analysing, and reporting qualitative research using thematic analysis [44]. Thematic analysis facilitated the emergence of themes and patterns, to identify broad concepts of the barriers and enabling factors that influence MCH nurses’ identification of risk of FV to First Nations women in Australia. This process included familiarisation with the data, then generating initial codes, searching for themes, reviewing the themes, defining and naming the themes, and producing a report with the themes found within the data. A key First Nations academic with experience in conducting research with First Nations mothers in ‘the first 1000 days’ of their child’s life, and the two site coordinators, were involved in the analysis of the data to ensure there was an Indigenous lens applied to the thematic analysis of the data. As a result, I have not accepted this recommendation.
Reviewer One: Summary: As a result, you can write what you found in relation to the purpose and what is your result next to the suggestion.
Author’s Response: I believe I have documented the results of the research in relation to the purpose and the implications for policy and practice in the original manuscript in line with Braun & Clarke’s six-step process for identifying, analysing, and reporting qualitative research using thematic analysis [44]. As a result, I have not accepted this recommendation.
Reviewer One: I also think that determining the 3rd title as "Matertal" or "Matertal and Method" would be more appropriate.
Author’s Response: Apologies, I don’t understand what you are suggesting here. I am presuming the word "Matertal", is meant to be “Maternal”? I can’t see where/what you are referring to. As a result, I have not accepted this recommendation.
Reviewer One: Also, the purpose and research questions should be at the end of the introduction, not under the method.
Author’s Response: I have been taught in my training in Australia to put the research questions where they are for this type of manuscript. I would prefer not to change where it is if that’s acceptable to you.
Reviewer One: I also think that there should be an ethics title (ethics committee and number) in the method section.
Author’s Response: Thanks for this feedback which I have accepted and edited the manuscript accordingly.
Reviewer 2 Report
Comments and Suggestions for Authors
Suggestions are in the attached file

Author Response
Author’s response to Reviewer Two’s comments
Reviewer Two’s comment:
The study contributes to the understanding of factors that influence maternal and child health nurses to identify the risk of family violence against Indigenous women in Australia. It is a relevant study, which shows the importance of research that addresses cultural aspects. Some adjustments are necessary for the study to be more consistent for publication:
- The sections are different from those included in the journal’s rules
Author’s response to Reviewer:
Thank you for your positive feedback. I have edited the relevant sections to align with the journal’s rules. Evidence of this can be seen in the tracked and clean copies of the revised manuscript.
Reviewer Two’s comment:
- Throughout the manuscript, the terms “woman with children aged birth to five years’’ and “MCH nurses” are repeated many times. In my opinion, if it is in the method that the study involved these two categories of interviewees, when citing women and nurses I see no need to repeat “woman with children aged birth to five years’’ and “MCH nurses” so many times.
Author’s response to Reviewer:
Thank you for your feedback. I have reviewed the manuscript and edited where appropriately. Evidence of this can be seen in the tracked and clean copies of the revised manuscript.
Reviewer Two’s comment:
- One of the biggest problems I identified is the presentation of the results (discussed further/responded to below)
Reviewer Two’s comment:
ABSTRACT
According to the journal’s rules, titles should not be included in the abstract sections.
Author’s response to Reviewer:
Thank you for this comment. I don’t believe that I used the title of the research in the abstract though.
Reviewer Two’s comment:
Furthermore, I found it strange that an “impact” section was included.
Author’s response to Reviewer:
The author has edited this section in response to your feedback.
Reviewer Two’s comment:
Regarding the results, it is not clear when the author states “ the nurses identified drugs, alcohol, socio-economic issues, history and stress…” History of what? I suggest specifying the history of what.
Author’s response to Reviewer:
Thanks for this feedback. The author has edited the Manuscript in response: “The nurses identified drugs, alcohol, socio-economic issues, the history of effects of colonisation on First Nations peoples, and stress as perceived factors influencing family violence, and acceptance, fear, cultural beliefs and mistrust, for women’s low reporting of violence”.
Reviewer Two’s comment:
INTRODUCTION and BACKGROUND
The journal’s rules do not include these two sections.
Author’s response to Reviewer:
Thank you for your feedback. I have edited the relevant sections to align with the journal’s rules. Evidence of this can be seen in the tracked and clean copies of the revised manuscript.
Reviewer Two’s comment:
The study’s problem is presented adequately. I suggest reviewing all the acronyms in the text and references to describe them before using them. Several acronyms are included without prior description.
Author’s response to Reviewer:
Thank you for your feedback. I have edited the manuscript in response to reviewing all the acronyms in the text and references to describe them before using them.
Reviewer Two’s comment:
THE STUDY
3.1 Aims: I suggest reviewing the objective because the second sentence of the objective paragraph is a justification for the study and not an objective.
Author’s response to Reviewer:
Thank you for your feedback. The author agrees and has edited the manuscript in response by removing this sentence.
Reviewer Two’s comment:
3.5 Sample/Participants: It was not clear to me how the sample size was determined. I suggest better detailing.
Author’s response to Reviewer:
Thank you for your feedback. I have edited the manuscript in response to make this clearer. Evidence of this can be seen in the tracked and clean copies of the revised manuscript.
Reviewer Two’s comment:
3.6.2 First Nations women with children aged birth to five years: Section presented with a very long and confusing paragraph. I suggest restructuring this paragraph.
Author’s response to Reviewer:
Thank you for your feedback. I have edited the manuscript in response to make this clearer. Evidence of this can be seen in the tracked and clean copies of the revised manuscript.
Reviewer Two’s comment:
FINDINGS
The results show the biggest problems:
- It is not coherent to include references in the results. Since the presentation of the results in this study are not included in the discussion, only the results found in this study should be included in this section.
Author’s response to Reviewer:
Thank you for your feedback, the author agrees and has edited this section of the manuscript in response. Evidence of this can be seen in the tracked and clean copies of the revised manuscript.
Reviewer Two’s comment:
- If it is a qualitative study, why put so many results in percentages? I don’t think it is necessary to put these numbers.
Author’s response to Reviewer:
Thank you for your feedback. As discussed in Real qualitative researchers do not count: The use of numbers in qualitative research - Sandelowski - 2001 - Research in Nursing & Health - Wiley Online Library, in quantitative research, numbers are used in qualitative research to establish the significance of a research project, to document what is known about a problem, and to describe a sample. But they are also useful for showcasing the labor and complexity of qualitative work and to generate meaning from qualitative data; to document, verify, and test researcher interpretations or conclusions. As such, I have not edited the manuscript as a result of this feedback.
Reviewer Two’s comment:
- Very long paragraphs, especially those on pages 10, 11 and 12. These are relevant results, but they need to be presented and described more appropriately to stimulate the reader. The way they are described, they are very confusing and discouraging. I suggest rewording them. Perhaps separate the nurses’ and women’s statements. The format of the text needs to be different.
Author’s response to Reviewer:
Thank you for your feedback, the author agrees and has edited this section of the manuscript in response. Evidence of this can be seen in the tracked and clean copies of the revised manuscript.
Reviewer Two’s comment:
DISCUSSION
In general, it is adequate, discussing the results with relevant and updated literature.
Page 15- Lines 576 to 588: this paragraph cites a review carried out in 2021, but the author was not cited.
Author’s response to Reviewer:
Thanks for this feedback, Citation now included:
- Austin, C.; Hills, D.; Cruickshank, M. Models and Interventions to Promote and Support Engagement of First Nations Women with Maternal and Child Health Services: An Integrative Literature Review. Children 2022, 9, 636. https://doi.org/10.3390/children9050636.
Reviewer Two’s comment:
Page 15- Lines 589 to 611: this paragraph also cites a study carried out in 2009, but the author was not cited.
Author’s response to Reviewer:
Thanks for this feedback, Citation now included:
- Victorian Aboriginal Community Controlled Health Organisation. Aboriginal families’ engagement in Maternal and Child Health services. Phase One: Project Report, 2015. Melbourne.
Reviewer Two’s comment:
5.2 Limitations
I didn’t understand why the second limitation “Second, due to the small number of participants, protecting anonymity may become an issue for those participants who do not want their stories shared”.
Generally, in qualitative studies the samples are small and, furthermore, the author estimated that data saturation would be achieved by recruiting ten participants in each category and subcategory.
Author’s response to Reviewer:
Thanks for this feedback. The author believes that this limitation is valid as although the author believes that by recruiting ten participants in each category and subcategory data saturation would be achieved, the study was conducted in a small rural community where anonymity is a very real limitation. As such, the author has not accepted this feedback.
Reviewer Two’s comment:
REFERENCES
I suggest reviewing all references in the text and in the list. On page 5, for example, the author’s name is separated from the year. And in the list, some references are mixed up, such as number 9, which has 2 references (Braun, V & Clarke, V. (2021) and Bryany, W., & Cussen, T. (2015).
Author’s response to Reviewer:
Thanks for this feedback. The author has done an extensive edit on the reference list and updated where necessary. Evidence of this can be seen in the tracked and clean copies of the revised manuscript.